# Is It Possible to Eradicate Carbapenem-Resistant *Acinetobacter baumannii* (CRAB) from Endemic Hospitals?

**DOI:** 10.3390/antibiotics11081015

**Published:** 2022-07-28

**Authors:** Filippo Medioli, Erica Bacca, Matteo Faltoni, Giulia Jole Burastero, Sara Volpi, Marianna Menozzi, Gabriella Orlando, Andrea Bedini, Erica Franceschini, Cristina Mussini, Marianna Meschiari

**Affiliations:** 1Department of Infectious Diseases, Azienda Ospedaliero-Universitaria of Modena, 41124 Modena, Italy; filippomedioli@gmail.com (F.M.); erica.bacca@gmail.com (E.B.); matteo.faltoni@gmail.com (M.F.); g.burastero@gmail.com (G.J.B.); saravolpi@outlook.com (S.V.); marymenozzi@gmail.com (M.M.); gabriella.orlando7@virgilio.it (G.O.); andreabedini@yahoo.com (A.B.); ericafranceschini0901@gmail.com (E.F.); 2Clinic of Infectious Diseases, Department of Infectious Diseases, University of Modena, 41124 Modena, Italy; crimuss@unimore.it

**Keywords:** gram-negative, *Acinetobacter baumannii*, carbapenemase, multi-drug resistance, carbapenem-resistant, outbreak, infection prevention, infection control, *Acinetobacter baumannii* carbapenemase, intensive care units

## Abstract

Background: Despite the global efforts to antagonize carbapenem-resistant *Acinetobacter baumannii* (CRAB) spreading, it remains an emerging threat with a related mortality exceeding 40% among critically ill patients. The purpose of this review is to provide evidence concerning the best infection prevention and control (IPC) strategies to fight CRAB spreading in endemic hospitals. Methods: The study was a critical review of the literature aiming to evaluate all available studies reporting IPC measures to control CRAB in ICU and outside ICU in both epidemic and endemic settings in the past 10 years. Results: Among the 12 included studies, the majority consisted of research reports of outbreaks mostly occurred in ICUs. The reported mortality reached 50%. Wide variability was observed related to the frequency of application of recommended CRAB IPC measures among the studies: environmental disinfection (100%); contact precautions (83%); cohorting staff and patients (75%); genotyping (66%); daily chlorhexidine baths (58%); active rectal screening (50%); closing or stopping admissions to the ward (33%). Conclusions: Despite effective control of CRAB spreading during the outbreaks, the IPC measures reported were heterogeneous and highly dependent on the different setting as well as on the structural characteristics of the wards. Reinforced ‘search and destroy’ strategies both on the environment and on the patient, proved to be the most effective measures for permanently eliminating CRAB spreading.

## 1. Introduction

Carbapenem-resistant *Acinetobacter baumannii* (CRAB) represents a major concern among carbapenem-resistant organisms (CRO) and is an emerging worldwide emergency. This pathogen finds its primary and exclusive spreading site into the healthcare setting [1,2].

According to the European Center for Disease Control (ECDC), CRAB has been identified as one of the 10 most frequently isolated microorganisms in ICU-acquired healthcare-associated infections (HAIs), accounting for 14.7% of pneumonia episodes, 8.1% of bloodstream infection (BSI), and for 7.1% of urinary tract infection (UTI) episodes in Italy [3]. The reported percentage of AMRI (Antimicrobial Resistance Index) associated with CRAB-ICU-acquired HAIs is 63.9% in Europe.

Such prevalence among HAI relates to the ability of *A. baumannii* to adhere to medical devices, including venous catheters (CVCs), urine catheters (CVs), and mechanical ventilation equipment, and its survival up to 33 days on dry surfaces [4,5]. Adhesion to various biotic and abiotic surfaces is the starting point for host colonization and infection. Biofilm bacteria are 10–1000 times more resistant to antibiotic treatment than the planktonic phenotype [4].

Several studies have been trying to investigate risk factors for colonization and infection with *A. baumannii* [6,7,8,9]. Unfortunately, most of these studies are heterogeneous and conducted in different epidemiological settings, with many different selection criteria between cases and controls, thus not allowing to extract conclusive results.

Latibeaudiere et al. showed, though, that previous CRAB colonization increased the risk to develop a CRAB infection eight-fold [7].

Recently a retrospective matched case–control [6] with a prospective inclusion of cases and concurrent selection of controls, demonstrated via multivariable analysis that significant risk factors associated with CRAB colonization were use of permanent devices, mechanical ventilation, McCabe score, and carbapenem use. Different risk factors have been related to different clinical contexts (geriatric department: UCs and CVCs, fatal comorbidity, longer length of hospital stay; internal medicine department: partial disabilities or bedridden status, prolonged hospitalization, previous admission to the ICU + MV, permanent devices and catheters, current antibiotic therapy or antibiotic polytherapy; ICU: high McCabe Score, use of t3GC and carbapenems) [10].

Moreover, upon exposure to antibiotic-based disinfectants, bacteria respond by forming a subpopulation that persists and can become highly tolerant to antibiotics. This ‘selected’ subpopulation plays an essential role in the lingering of biofilm infections.

Moreover, the acquiring of carbapenem resistance leads to limited therapeutic options and this is linked with a high rate of mortality. Several strains developed the capacity to transmit resistance via mobile genetic elements that enable the production of carbapenemase enzymes [11]. This enhances the burst of outbreaks in healthcare settings. Thus, implementing a correct infection prevention and control (IPC) policy, involving all the healthcare professionals altogether, is essential.

CRAB outbreaks have mainly been reported in ICUs, during mechanical ventilation, after antibiotic treatment, and showed a higher mortality rate than *Pseudomonas aeruginosa* (47% vs. 23%) [7].

An important role has been played by the pandemic, since the latest European Antimicrobial Resistance Surveillance Network (EARS-Net) investigating the impact of the COVID-19 pandemic on antimicrobial resistance (AMR) underlines a decreasing trend of co-infection due to community pathogens in COVID-19 in-patients, in contrast with an excess of MDROs responsible for COVID-19 superinfections [12,13,14,15].

In particular, CRAB and vancomycin-resistant *E. faecium* were isolated more frequently in 2020 than in the previous years.

This event may be related to the fact that infection prevention and control (IPC) and antimicrobial stewardship (AS) programs have been compromised during the pandemic, leading to hospital-onset MDRO outbreaks [16,17,18,19,20].

In this emergent scenario, it is essential to better define which IPC interventions is relevant to eliminate CRAB from hospitals and should be considered an absolute priority. In general, a multimodal IPC approach, better if implemented as a ‘bundle’ of interventions, has been proven to be more effective [21,22,23].

In 2017, both ECDC and WHO guidelines were published in order to address this important issue.

The ECDC and WHO rigorously performed a systematic literature review to identify the best available evidence on the effective IPC measures to be applied for all at-risk patients upon admission to healthcare settings to prevent the transmission of all carbapenem-resistant organisms (CROs), including CRAB. The WHO document offers important additional suggestions for best practices to turn recommendations into adaptive work. 

However, in both these guidelines, the quality of evidence on CRAB control were lowered from low to very low due to the scarce number and poor quality of the studies included. The described measures varied significantly in scope and evidence based [8,22]. 

Moreover, studies published after 2016 were obviously not included. 

The consequences of this uncertainty are that it is still unclear which could be the best approach, especially when resources are limited, or times are challenging (like those of the COVID-19 pandemic). 

The aim of this critical literature review is to identify the most effective IPC strategy to face the rising problem of CRAB spreading in hospitals worldwide.

## 2. Materials and Methods

We performed a critical literature review to assess published evidence on the control and prevention of CRAB in ICU during the last 10 years. As a first step, we decided to focus our review method on the infection control (IC) measures used for evidence grading in the European Society of Clinical Microbiology and Infectious Diseases (ESCMID) guidelines for the infection control measures to reduce transmission of multidrug-resistant Gram-negative Bacilli (GNB), published in 2014 [21]. 

Hence, publications from 2012 to 2022 were outlined via an online literature search using PubMed. 

We used the search criteria described below: *Acinetobacter AND CRAB OR Acinetobacter baumannii OR carbapenem-resistant *Acinetobacter baumannii*) AND (cross infection OR infection control OR infection prevention OR patient isolation OR cohorting OR gloves OR protective clothing OR handwashing OR hand hygiene OR sanitizer OR cleanser OR disinfectant OR pre emptive isolation OR antisepsis OR disinfection OR sterilization OR environmental cleaning OR screening culture OR disease outbreaks OR management*. Filters: from 2012–2022.

The articles extracted were then audited for meeting inclusion or exclusion criteria. 

Inclusion Criteria:The paper was published, but only full articles;The paper was an epidemiological and/or outbreak report (both in endemic and in epidemic setting);The responsible agent of the outbreak was CRAB;The paper included description and assessment of IC measures deployed during the outbreak;The paper included incidence and/or prevalence of the CRAB infection/colonization;The outcome of the outbreak was described.

Exclusion Criteria:In vitro data;All other carbapenemases;Only diagnostic data;Case reports;Reviews;Other species than human.

The gathered IC measures were graded into major topics, based on the recommended measures to reduce transmission of CRAB from international agencies or international professional societies, i.e., the World Health Organization, the European Centre for Disease Prevention and Control, the U.S. Centers for Disease Control and Prevention, the U.S. Agency for Healthcare Research and Quality, and the European Society of Clinical Microbiology and Infectious Diseases [8,21,22,24,25].

The recorded measures consisted of hand hygiene; alcohol hand rub consumption; active rectal screening; additional active screening strategies; contact precautions and room isolation; alert code; daily chlorhexidine baths; staff/patient cohorting; closure/stop admissions; environmental disinfection; environmental cultures; monitoring of environmental cleaning; genotyping; and antimicrobial stewardship/monitoring of antibiotics consumption; training/education. The infection control studies were distinguished based on country origin, type of study, hospital setting, and department involved. The statistical method and final study outcomes were also assessed. 

The outcome has been defined as a reduction in CRAB isolation during the reported period after the implementation of the mentioned IPC measures (as described in the ECDC guidelines, both for endemic and epidemic settings) [21].

## 3. Results

Our initial search query revealed 254 records, of which 58 met the inclusion criteria and were evaluated by full text. Finally, 12 articles were included that reported IPC measures for CRAB.

### 3.1. Study Characteristics

Table 1 summarizes the characteristics of the 12 included studies. The described studies and outbreaks reported data from all continents, supporting the potential for endemic spread of CRAB. Most of the studies described an epidemic outbreak (58%) while 42% occurred in endemic settings. The total duration of these studies was very heterogenous, ranging from 3 months to 7 years 

The majority of the studies were performed in an ICU (83%). Four studies included data from outside an ICU setting [20,27,28,29]. None of the studies were set exclusively outside of an ICU. Regarding the ICU structure, only 5 out of 12 were ‘closed’ ICUs. 

Concerning the statistical analysis conducted: half of the studies used an interrupted time-series analysis (ITS); three used a segmented regression analysis (SR); 1 a before and after study (BA); one used a cross-sectional study (CS); one used a joinpoint regression analysis (JP), even though this was combined with an ITS; and one was an intervention time series analysis (ITSA). Considering intervention outcomes: all except one study reported a significant reduction in CRAB post-intervention incidence. The duration of follow-up after the intervention ranged from 0 months (interrupting the observation at the end of the intervention) to 3 years after the intervention.

### 3.2. IPC Measures 

Table 2 describes the most frequent components in infection prevention and control multimodal interventions implemented and their relative positive/negative impact on the outcome. All the studies adopted a multimodal approach with more than five different combined interventions.

The implemented IPC measures were: environmental disinfection (100%) more frequently performed with 10% sodium hypochlorite; hand hygiene and/or alcohol-based hand rub consumption (91%); contact precautions (83%); staff education (83%); additional active screening (83%); cohorting staff and patients (75%); monitoring of environmental cleaning (66%); genotyping (66%); daily chlorhexidine baths (58%); antimicrobial stewardship/monitoring of antibiotic consumption (58%); active rectal screening (50%); environmental cultures (41%); and closing or stopping admissions to the ward (33%).

Contact precaution was considered an essential component for CRAB control and was universally applied by all studies, while the use of single rooms or rather than enhanced cohorting using a separate intensive care module varied across studies. One study adopted universal contact precaution until the patient was discharged independently of CRAB status. Overall, 9 out of 12 of the studies adopted a cohorting strategy (2 did not mention such strategy).

Staff or nursing cohorting was mentioned in three of the studies [29,30,31].

Enhanced training and staff education was achieved in all except two studies. However, many studies did not specify the implementation model.

Unexpectedly, the implementation of hand hygiene best practices was described in only eight of the studies, and alcohol hand rub consumption was described only in five of the studies.

Only three of the studies, in fact, thoroughly depicted the implementation methodology they used.

More specifically, Cho et al. reported the promotion of hand hygiene using alcohol-based hand gel (ABHG) without any further description or registration of the intervention [26].

Chung et al. stated that they followed the current guidelines according to hand hygiene [32], without subsequent implementation.

Valencia-Martin et al. provided both training and structured observation on hand hygiene, without focusing on alcohol-based hand rub consumption [28].

Enfield et al. measured hand hygiene compliance through a covert, observation program that has been used at UVAMC since 2006 (no further description was provided) [29].

Munoz-price et al., on the other hand, described how, in their intervention, hand imprints were sporadically obtained from the staff. The plates with bacterial growth were then returned to the units and shown to health professionals to explain the potential role of their hands in the spreading of bacteria. Moreover, positive plates were published as examples in the weekly electronic communications. Finally, hand hygiene messages were placed via posters with pictures of hospital leadership personnel [27].

Active surveillance screening was widely varied depending on the surveillance site, the number of samples, and their frequency. Seven out of twelve adopted rectal screening, three out of thirteen (Metan, Meschiari, and Eckardt) added systematically the screening of axilla and groin; Perez also added the respiratory tract samples but with a random recurrence. Moreover, Valencia-Martìn et al. were able to reach a sensitivity of 96% combining rectal and pharyngeal swabs, compared to the 78% obtained with rectal swab only. Combining the overall results of the studies, the best performance has been obtained with skin samples (100%), followed by rectal samples (86%) [29]. The frequency of repeated screenings was also variable: half of the studies, screened actively, starting from ICU admission then repeated once a week. For the others, screening ranged from twice per week to once every two weeks.

Whole-genome sequencing analysis (WGS) was applied only by 25% of studies.

## 4. Discussion

The literature, regarding CRAB, contains information that are extremely variable from many different settings worldwide. Therefore, even after a thorough selection of articles, it remains difficult to describe a pattern of interventions that could be universally effective. Nonetheless, we were able to identify the most used and the most effective measures against CRAB in those that we deemed to be the most representative experiences of the last 10 years. Furthermore, by including very different studies in terms of setting (both endemic and epidemic) and country of origin, our study allows us to validate the effectiveness of interventions in geographical areas that differ widely in terms of incidence rates and availability of resources.

Indeed, the CRAB prevalence in the mentioned countries varies as follows: USA 30% (25–35%); South Korea 77% (71–82%); Greece 94% (92–95%); Spain 58% (47–68%); China 82% (80–84%); Turkey 91% (90–92%); and Italy 80% (78–82%) [37,38]. Therefore, the wide variability gave us the possibility to describe how to control CRAB spreading in those countries for which the most are epidemiologically affected by this pathogen [39].

Concerning the statistical method used to define the results, only half of the studies used an interrupted time-series analysis to evaluate their outcomes and only one study used an intervention time-series analysis that demonstrated that an ICP bundle including enhanced environmental cleaning had a decisive impact on nosocomial CRAB ICU incidence density against a background of stable AHR and antibiotic use [33].

Importantly, only these analyses could provide an appropriate evaluation of ICP measures because it provides an overview of the well-distributed effectiveness over time, enriching the assessment methodology with quality [11,40].

Furthermore, to provide evidence of the sustainability of the intervention performed, it is important to use a long post-intervention follow-up. Most of the studies included (9 out of 12) measured their interventions during a time lapse that ranged from 1 year to 7 years [26,27,28,29,30,32,33,34,35].

Another relevant consideration is that the CRAB burden seems to affect the intensive care setting more significantly; only a few studies have addressed this issue outside ICU. Further studies would be required to investigate the burden of this pathogen in non-intensive areas and to demonstrate whether the proposed interventions could be equally effective.

Importantly, the impact of ICP measures in ICUs could also be influenced by the open versus closed structure.

There is good evidence that closed ICUs are associated with better outcomes and better quality of care, other than being less prone to wide-spread infections by MDROs, indeed, closed ICUs allow easier implementation of contact precautions and cohorting of patients than open areas.

However, as our review shows, these formats are increasing lacking, and the implementation of ICP strategies must deal with these structural limitations. Strategies to control CRAB outbreaks often require positive patients’ relocation to a cohort ward and sometimes even lead to temporary closure of the ICU. Therefore, contact isolation, which was confirmed as a key strategy for CRAB control, implemented in all the studies reviewed, cannot always be performed by placing CRAB colonized/infected patients in a single room. Similarly, patient cohorting is extremely difficult to apply in open space units. On the other hand, it is equally impossible to close ICU if it is the only one in the hospital or during an epidemic period such as the recent COVID-19 pandemic.

In countries with limited resources and unfavorable structures, other than in pandemic periods, Meschiari et al. suggested, wherever the setting is an open space, innovative solutions such as cycling radical cleaning and disinfection. This procedure can be easily implemented in open-space ICUs and avoids ICU closure and limited admissions.

Contact isolation is strictly linked to active surveillance, applied in the 83% of the revised studies. Active screening for early detection and control of CRAB, while strongly recommended for carbapenem-resistant Enterobacteriaceae (CRE), is still strongly debated.

For instance, ECDC guidelines suggest applying active screening, as for CRE, by obtaining swabs from rectal or perirectal areas, and any other site that is either actively infected or considered to be colonized [32]; on the other hand, ESCMID guidelines underline that the detection of a CRAB carrier may be affected by the low sensitivity of the conventional methods. The WHO guidelines did not address this important topic [29]. Only Tacconelli et al., quoting the Association for Professionals in Infection Control and Epidemiology (APIC) guide for the control of MDR-*A. baumannii*, suggested to culture multiple patient sites including the nose, throat, axilla, groin, rectum, open wounds and/or tracheal aspirates [21,41].

Accordingly, to these heterogenous recommendations, our review confirmed different approaches of active screening at country level.

Our findings suggested that, for endemic situations, it was sufficient to maintain active surveillance within 48 h of ICU admission with a rectal swab; during outbreaks, surveillance must be implemented weekly, granted by rectal swabs, and adding at least three other different body areas (axilla, groin, and trachea were specifically the most used).

It is important to note that since the results of the CRAB screening are not immediately available, and that the timing of application of the different measures depends on the method used (molecular PCR or phenotypical culture), our previous work suggested to apply preventive contact precautions to all the ICU patients until the end of the outbreak [24].

Unfortunately, the implementation of hand hygiene best practices was described in only eight of the studies, and alcohol hand rub consumption data were collected only in five of the studies.

Effective hand hygiene compliance is widely recognized and strongly recommended, both by the WHO and ECDC [8,22], to reduce healthcare-associated infection transmission; thus, it should be a standard of care and is not outlined among interventions [11]. Moreover, it seems crucial, especially during outbreaks, to conduct specific audits and feedback on hand hygiene direct compliance, which is better if in the field, and only one study seemed to use this approach [42]. Alcohol-based hand rub (ABHR) consumption is depicted as a valid alternative and/or addition to hand hygiene implementation in the guidelines, even though according to our review, only a few studies reported the percentage of ABHR use during the study period.

Unexpectedly, despite CRAB being well-known for being strongly environmentally resistant as well as a biofilm producer, environmental screening, and subsequent disinfection of colonized surfaces, it was described only in half of the studies reviewed. Its role is extremely debated, and only conditionally recommended [22] (only WHO guidelines mention environmental screening), because the traditional method of environmental sampling suffers from low sensitivity and requires an important consumption of human resources, other than a massive cooperation with the microbiology laboratory.

To overcome the limitation due to poor sensitivity, Meschiari et al. used a brain–heart infusion (BHI) moistened sterile gauze technique because in their experience it proved to be far more sensitive than standard sampling methods (40% positives vs. 0%; *p* < 0.05). More than 50% of the environmental samples in that study were positive for CRAB [38], while instead, the other studies that gathered environmental sampling did not describe the use of such a methodology.

On the other hand, 100% of the studies performed environmental cleaning, confirming its crucial role for CRAB eradication. In this regard, the disinfectant used did not seem to be essential (9 out of 12 (75%) of the studies used sodium hypochlorite), but rather the certainty of the cleaning and biofilm complete removal.

The evidence that seemed to support the use of new technologies, such as the peroxide or the UV rays, was recently disproved [43,44,45].

Concerning patient decolonization, the evidence for effectiveness of chlorhexidine gluconate (CHG) bathing against CRAB is a relatively recent finding. This could be the reason why only seven of the studies utilized daily chlorhexidine baths for the patients. The 2014 ESCMID guidelines did not recommend the universal use of chlorhexidine because of a lack of evidence concerning the reduction in bloodstream infections due to Gram-negative bacilli [21]. Nevertheless, they mentioned some successful bundles that included chlorhexidine baths.

Concerning the importance of implementation of training and education and the composition of the infection control team, unfortunately only three of the studies went into detail about what professionals were included into the intervention (namely nurses, ICU physicians, infectious disease physicians, and microbiologists). Further studies about this matter may be required [27,28,33].

A recent study by Fan et al. [46] suggests that chlorhexidine bathing significantly reduces CRAB colonization in the ICU setting.

To provide useful information on the CRAB transmission dynamic, only some of the studies used advanced genotyping methods (66%), both for patient and environmental screening. Unfortunately, next-generation sequencing was applied only by 25% of the studies included. Specifically, our finding underlined that only WGS proved to be a valuable tool for identification of the sustained reservoirs.

The issue of therapy to control CRAB must be addressed as selective pressure and carbapenem sparing [47]. For this very reason, 7 out of 12 of the papers reviewed focused on antimicrobial stewardship, and especially focused on the carbapenem sparing part, reducing, in all cases, the prevalence of CRAB. It seems then that of vital importance in involving infectious diseases, physicians, microbiologists, and pharmacologists into a stewardship program, to decrease the resistance pressure on colonizing *Acinetobacter baumannii*.

These results emphasize how it will be essential for the future to invest economical resources to reach more profitable results.

Our critical review certainly has limitations; it is not a systematic review, and we used only one database for the research (PubMed).

Moreover, due both to the limited number of studies published addressing this issue, and their extreme variability in terms of methodology, study design, setting, outcome, and duration of follow-up after intervention, it was impossible to obtain a direct comparison as well as the possibility of publication bias demonstrating the effectiveness of certain IPC measures in CRAB hospital eradication. This should be considered in view of the included studies.

## 5. Conclusions

The impact of every single IC measure against the spreading of CRAB remains difficult to assess. The quality of the evidence published so far is still low, and there is a lack of controlled intervention studies.

Even if variability in outcomes and measures, along the studies, is still wide, the implementation of multimodal measures achieved a significant reduction in CRAB infection and CRAB-related deaths. Reinforced ‘search and destroy’ strategies both on the environment and on the patient, proved to be the most effective measures for permanently eliminating CRAB spreading.

Our results underline how intervention bundles should be coherent with the setting where they are applied. The COVID-19 pandemic demonstrated that open-space ICUs promoted the transmission of pathogens with greater environmental resistance, such as CRAB. Therefore, it is of vital importance to develop strategies that allow to maintain open wards and do no limit access, either in ICUs or any other ward in order to overcome nosocomial outbreaks without limiting or even stopping healthcare activities.

## Figures and Tables

**Table 1 antibiotics-11-01015-t001:** Characteristics of the studies included.

Study	Country	Type of Study	Setting	Department Involved	ICU y/n	Statistical Method
Perez et al., 2020 [19]	USA	Outbreak Report	Epidemic	Acute Care Hospital	No	ITS: CRAB_ID decreased
Cho et al., 2014 [26]	South Korea	Research support, Non-U.S. Gov’T	Endemic	Tertiary Hospital	No	SR: CRAB_ID decreased
Munoz-Price et al., 2014 [27]	USA	Major Article	Endemic	Major Hospital	Yes	SR: CRAB hospital acquired cases decrease
Valencia-Martìn et al., 2019 [28]	Spain	Research	Endemic	Intensive care Unit + adult wards	Yes	JP + ITS: SC decreased
Enfield et al., 2014 [29]	USA	Comparative Study	Epidemic	Intensive care Unit	Yes	ITS: CRAB_ID decreased
Karampatakis et al., 2018 [30]	Greece	Epidemiological Stuy	Endemic	Intensive care Unit	Yes	ITS: linear trend of CMA for CRAB infections increased
Eckardt et al., 2022 [31]	USA	Major Article	Epidemic	Intensive care Unit	Yes	ITS: CRAB_ID decreased
Chung et al., 2015 [32]	South Korea	Research support, Non-U.S. Gov’T	Endemic	Intensive care Unit	Yes	ITS: CRAB_ID decreased
Meschiari et al., 2020 [33]	Italy	Research	Epidemic	Intensive care Unit	Yes	ITSA: CRAB_ID decreased
Zhao et al., 2019 [34]	China	Research	Epidemic	Intensive care Unit	Yes	CS: prevalence of CRAB decreased
Ben-chetrit et al. [35]	Israel	Research	Epidemic	Intensive care Unit	Yes	SR: CRAB_ID decreased
Metan et al., 2019 [36]	Turkey	Original Article	Epidemic	Neurological Intensive care Unit	Yes	BA: CRAB_ID decreased

CRAB_ID CRAB, incidence density; CS, cross-sectional study; ICU, intensive care unit; ITS, interrupted time-series analysis; ITSA, intervention time series analysis; JP, joinpoint regression analysis; LC, linear change; SC, slope change; SR, segmented regression analysis; CMA, centered moving average; BA, before and after study.

**Table 2 antibiotics-11-01015-t002:** Most frequent components in infection prevention and control multimodal interventions implemented in the studies included.

Study	HH Compliance/AHR Consumption	Active Rectal Screening (Targeted/Universal)	Additional Active Screening Strategies	Contact Isolation /Alert Code	Daily Chlorhexidine Baths	Cohorting Staff/patients	Closure/Stop Admissions	Environmental Disinfection	Environmental Cultures	Monitoring of Environmental Cleaning	Genotyping	Antimicrobial Stewardship/Monitoring of Antibiotic Consumption	Training /Education	Outcome
Perez et al., 2020 [19]														
Cho et al., 2014 [26]														
Munoz-Price et al., 2014 [27]														
Valencia-Martìn et al., 2019 [28]														
Enfield et al., 2014 [29]														
Karampatakis et al., 2018 [30]														
Eckardt et al., 2022 [31]														
Chung et al., 2015 [32]														
Meschiari et al., 2020 [33]														
Zhao et al., 2019 [34]														
Ben-chetrit et al. [35]														
Metan et al., 2019 [36]														
All studies														
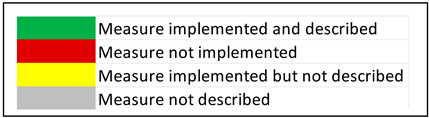														

Legend: AHR, alcohol-based hand rubs; BA, before and after study; BHI, brain–heart infusion medium; CMA, centered moving average; CRAB, carbapenem-resistant *Acinetobaster baumannii*; CRAB_ID CRAB, incidence density; CS, cross-sectional study; DDD, defined daily doses; ERIC-PCR, Enterobacterial repetitive intergenic consensus; HH, hand hygiene; ICU, intensive care unit; NA, not available; PGFE, pulsed-field gel electrophoresis; BA, before and after analysis; WGS, whole-genome sequencing.

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
