# Peer review of "Is It Possible to Eradicate Carbapenem-Resistant *Acinetobacter baumannii* (CRAB) from Endemic Hospitals?"

_antibiotics, 2022, doi:10.3390/antibiotics11081015_

Round 1

Reviewer 1 Report

The reviewed work is good and interestinga but I found different errors in words that must be in Italic style or not, and throughout the entire manuscript it is necessary an English revision. Moreover, Table 2 is too small and Table 1 must be reviewed since I think that the reported data are not all important. At last, it is necessary to review the order of references in the text, in the introduction the first references start with 37 and 38.

Author Response

Thank you very much for your precious comment to our work. We thoroughly reviewed English language and we looked through for Italics misspellings.

We also reviewed our Table 2, according to your suggestion, to make it more readable, while, in our opinion, the variable included in Table 1 are all relevant to highlight the type of study, the source/setting and country of origin and the reliability/reproducibility of the results described in the following table (Table 2).

We also rearranged the references.

Thank you again for your suggestions.

Reviewer 2 Report

I read, Is it possible to eradicate carbapenem-resistant Acinetobacter baumannii (CRAB) from endemic hospitals? with interest. In this manuscript, the authors reviewed the evidence concerning the best infection prevention and control (IPC) strategies to fight CRAB spreading in endemic hospitals.

I have some questions and suggestions.

1. Can you explain why this review is new or telling new things?

2. Why did you use only one database (Pubmed) to search this review?

3. Discussion is rather weak. The data from other studies is relatively small. Please add more data about the effective IPC strategy to face the rising problem of CRAB spreading in hospitals from the other studies as well for comparison with this study.

4. Could you please insert/use/discuss the issue of treatment of CRAB infection? Please see,

1. https://doi.org/10.3390/pharmaceutics14010031

2. https://doi.org/10.3390/pharmaceutics14061266

5. Line 309-317: In addition to the guidelines, please add the original article to support this information.

6. Please add more limitations in your review. For example, use only one database

7. Please provide more data on the importance of physicians, pharmacists, and healthcare professionals around the world recognizing the infection prevention and control (IPC) strategies to fight CRAB spreading in endemic hospitals.

Minor

1. Line 308, 341, 348, 371,373: please add references.

Author Response

I read, is it possible to eradicate carbapenem-resistant Acinetobacter baumannii (CRAB) from endemic hospitals? with interest. In this manuscript, the authors reviewed the evidence concerning the best infection prevention and control (IPC) strategies to fight CRAB spreading in endemic hospitals.

We are very grateful for your constructive comments and suggestions to our paper entitled: Is it possible to eradicate carbapenem-resistant Acinetobacter baumannii (CRAB) from endemic hospitals?

We here provide a point-by-point reply to the comments, and we have incorporated the related changes in the manuscript (in yellow). We thank the reviewer for his thoughtful insights which helped to significantly improve the manuscript.

  1. Can you explain why this review is new or telling new things?

Thank you very much for this question: it grants us the possibility to underline the importance of our work. We deemed that our review would gather successful and updated experiences worldwide, especially in the current situation where the IPC guidelines provide weak evidence and that the pandemic has put pressure on all the healthcare settings, especially intensive care. In literature there’s no consensus concerning this matter so far, in particular due to limited data and well-designed studies on CRAB before 2017. Our review, including papers recently published (also during the recent pandemic) summarizes key strategies and aims to be a practical tool (guide) for clinicians who are facing CRAB epidemic in their hospitals.  To our knowledge an up-to-date review focusing only on CRAB nosocomial control is currently lacking.

  1. Why did you use only one database (Pubmed) to search this review?

 Thanks for the question. We decided to use only PubMed as after a selection of various databases because we deemed it would have been the most useful for our research, without polluting excessively our research with unrequited papers.

Namely:

 - Cochrane would have provided a more treatment-oriented information that it wouldn’t have been useful for our research

- TripPro would have been too clinically oriented

- EMBASE would have been too drug oriented

  1. Discussion is rather weak. The data from other studies is relatively small. Please add more data about the effective IPC strategy to face the rising problem of CRAB spreading in hospitals from the other studies as well for comparison with this study.

Thanks for the kind comment. Even though, we deem to have carried out in the discussion part all of the principal IPC interventions against CRAB, comparing the most substantial available studies in literature. Unfortunately, there aren’t many other studies that confront this issue. As we stated in our conclusions, IPC measures against CRAB are still quite an unexplored matter, that requires further and more solid evidence.

  1. Could you please insert/use/discuss the issue of treatment of CRAB infection? Please see,
  2. https://doi.org/10.3390/pharmaceutics14010031
  3. https://doi.org/10.3390/pharmaceutics14061266

We are deeply grateful both for the suggestion and for providing us with such interesting materials. Concerning the study design, though, since the very beginning we set the aim to discuss only about the infection control and prevention measures available against Carbapenem-resistant Acinetobacter baumannii, without dealing with the treatment issue. In our opinion, discussing about the treatment would represent a distracting element for our purpose. We will nonetheless read thoroughly the papers you sent us and include them in future studies. We really think that the best treatment for CRAB infections remains a matter of intense debate in the current literature and deserves a separate paper.

We also decided to add a sentence in the discussion (line 374-375).

  1. Line 309-317: In addition to the guidelines, please add the original article to support this information.

Thank you very much, you are right. We added the proper quote right away.

  1. Please add more limitations in your review. For example, use only one database

You are right, thanks for the observation. We right away added this limitation in the proper section [line 376 and on]

  1. Please provide more data on the importance of physicians, pharmacists, and healthcare professionals around the world recognizing the infection prevention and control (IPC) strategies to fight CRAB spreading in endemic hospitals.

This is a really useful information. We added for sure data about the importance of the implementation of all healthcare professionals into the Infection control process. However, many studies described the implementation and training of health personnel generically without going into detail so we could not analyze this important issue in the discussion section.

Nonetheless we added a section into discussion regarding this matter (line 367-371)